# Biomarker Development in Cardiology: Reviewing the Past to Inform the Future

**DOI:** 10.3390/cells11030588

**Published:** 2022-02-08

**Authors:** Katharine A. Kott, Michael Bishop, Christina H. J. Yang, Toby M. Plasto, Daniel C. Cheng, Adam I. Kaplan, Louise Cullen, David S. Celermajer, Peter J. Meikle, Stephen T. Vernon, Gemma A. Figtree

**Affiliations:** 1Cardiovascular Discovery Group, Kolling Institute of Medical Research, University of Sydney, St Leonards 2065, Australia; katharine.kott@sydney.edu.au (K.A.K.); steve.vernon@gmail.com (S.T.V.); 2Department of Cardiology, Royal North Shore Hospital, St Leonards 2065, Australia; 3Sydney Medical School, University of Sydney, Camperdown 2050, Australia; hyan2234@uni.sydney.edu.au (C.H.J.Y.); tpla0842@uni.sydney.edu.au (T.M.P.); dche9126@uni.sydney.edu.au (D.C.C.); akap9401@uni.sydney.edu.au (A.I.K.); david.celermajer@health.nsw.gov.au (D.S.C.); 4School of Medicine and Public Health, University of Newcastle, Kensington 2033, Australia; michael.bishop@uon.edu.au; 5Emergency and Trauma Centre, Royal Brisbane and Women’s Hospital, Herston 4029, Australia; louise.cullen@health.qld.gov.au; 6Department of Cardiology, Royal Prince Alfred Hospital, Camperdown 2050, Australia; 7The Heart Research Institute, Newtown 2042, Australia; 8Baker Heart and Diabetes Institute, Melbourne 3004, Australia; peter.meikle@baker.edu.au

**Keywords:** cardiology, biomarker development, atherosclerosis, heart failure, acute coronary syndrome, statistical development

## Abstract

Cardiac biomarkers have become pivotal to the clinical practice of cardiology, but there remains much to discover that could benefit cardiology patients. We review the discovery of key protein biomarkers in the fields of acute coronary syndrome, heart failure, and atherosclerosis, giving an overview of the populations they were studied in and the statistics that were used to validate them. We review statistical approaches that are currently in use to assess new biomarkers and overview a framework for biomarker discovery and evaluation that could be incorporated into clinical trials to evaluate cardiovascular outcomes in the future.

## 1. Introduction

The practice of modern cardiology utilizes multiple biomarkers and is an exemplar of biomarker discovery and use in medicine. While the drive to look for new and improved markers to benefit patients with cardiac disease is warranted, much research in this area over the past decade has had little impact on clinical care. Here, we review the development of some of the major cardiac biomarkers and discuss how they were first identified, what populations they were assessed in, and how their relationships to the conditions of interest were quantified. We have limited the scope of this review to protein-based markers, though similar approaches can and have been used with the extremely important lipid-based markers. Finally, we consider the future direction of biomarkers with new discovery platforms in well-phenotyped cohorts and the areas of remaining unmet need.

## 2. Statistical Approaches to Assessing New Biomarkers

Scientific discovery is the first step in biomarker development. Once a marker has been identified, characterized, and measured in the population of interest, statistically robust and clinically useful associations between the marker and the disease need to be established. If a significant association persists after adjustment for age and other established risk factors, it is often conveyed in relation to those with levels above a threshold as a relative risk in some form, such as a rate ratio, odds ratio, or hazard ratio [1]. Ultimately, the most compelling evidence comes from well-designed prospective studies demonstrating clinical utility [2], but this is not always performed prior to adoption in guidelines and practice [3].

Biomarker studies in cardiac disease have primarily used dichotomous outcomes, such as whether a person died or not, had a myocardial infarction or not, or had heart failure or not. This type of binary analysis generally reflects disease incidence rather than disease severity. Logistic regression analyses are used to evaluate the association of a biomarker with a dichotomous outcome. The distribution of the biomarker needs to be assessed for both cases and non-cases. If it is not normally distributed, transformation (commonly log2) frequently results in a sufficiently normal distribution to allow for analysis. Alternatively, if transformation is undesirable, or if the transformed data remain non-parametric, the biomarker can be categorized instead, commonly into quartiles, tertiles, or via a biologically plausible cut-off [4].

While some markers discussed below have been correlated with disease severity, the predictive utility of most of the biomarkers discussed in this article have been assessed using ROC curves, which are a visual representation of how well a model categorizes cases and non-cases. ROC curves have gained popularity in clinical science due to their intuitive interpretation and lack of dependency on units, granting the ability to make comparisons between different biomarkers or risk scores [4]. The area under the ROC curve, or the C-statistic, is equivalent to the likelihood that a randomly selected case is correctly given a higher rating or rank than a randomly selected non-case [5]. Its value ranges from 0.5 (no discrimination) to 1 (perfect discrimination) [6]. The C-statistic is a function of sensitivity and specificity, but it has limitations in clinical practice.

Clinically, we are often more interested in risk prediction, which is not captured by the C-statistic. The predictive value, or the post-test probability, is the likelihood of disease in patients with a positive test result [6]. This may be a more relevant measure, as it quantifies the likelihood of having the disease given the test result [6]. In cardiovascular medicine, we commonly use models which incorporate multiple variables (e.g., lipid levels, hypertension, and smoking) which would not result in a significant improvement in the ROC curve when assessed individually. The inclusion of novel biomarkers in risk prediction equations may not drastically alter the C-statistic; however, even incremental improvements can result in more accurate risk categorization and subsequent treatment.

Though evidence of association is a key step, it does not necessarily convey the clinical utility—or lack thereof—of measuring the new biomarker. This requires subsequent evaluation, usually in comparison with an existing model. Comparison between models involves assessment of the global measures of model fit, as well as measures of calibration, reclassification, and discrimination [7]. Calibration is the extent to which predicted probabilities agree with observed risk [6], and this can be assessed using the Hosmer–Lemeshow statistic [8,9], calibration-in-the-large, and a calibration plot [10]. Put simply, it is the frequency of which the estimation by the model reflects the real outcome [2]. Reclassification indicates whether the new model changes the estimate of an individual’s risk enough to reallocate them to a different risk category [7]. Discrimination is the extent to which a model can separate cases and non-cases (i.e., those who have the disease and those who do not). Discrimination is most useful when dichotomous categorization is the goal, such as in diagnostic testing, and is usually represented by the area under the receiver operating characteristic (ROC) curve, or the C-statistic [5,6].

In recent decades, novel ways to assess improvement in risk prediction have been proposed. Assessing reclassification has become a popular way to assess novel markers and models, but it has limitations [11]. For example, a reclassification when a novel factor is included in a risk model may be accurate and lead to more appropriate treatment for some patients, but it may also incorrectly reclassify some patients who will derive no benefit from this reclassification. For this reason, net reclassification improvement (NRI) [11] was developed to reward correct reclassification and penalize incorrect reclassification [4]. Extending this further, integrated discrimination improvement (IDI) is the difference between the integral of sensitivity and the integral of one minus specificity over all possible cut-off values [12]. Like the ROC, the IDI is a measure of the corrected average sensitivity. Its utility differs from the NRI, as it does not require risk categories for reclassification, which is useful when established risk cut-offs do not exist [11].

Another approach to evaluating the utility of novel biomarkers is decision tree analysis modeling [3,13]. This approach may be more appropriate when the goal is to analyze all logical diagnostic and treatment strategies in population subgroups, which are often categorized using already established cut-offs [3]. Decision tree analysis modeling allows for the comparison of multiple strategies, which may be more appropriate when optimizing models that use multiple variables. A recent meta-analysis used decision tree analysis to predict the severity and mortality in COVID-19 patients using known biomarkers [13], and interestingly, the best model incorporated age, troponin, and aspartate aminotransferase. However, further research is needed with prospective studies of this type of modeling. An overview of an integrated approach to biomarker discovery is presented in Figure 1.

## 3. The History of Biomarker Development in Cardiovascular Disease

The earliest reports of success of blood-based biomarkers providing clinical utility for cardiovascular disease were focused on improving the diagnosis of myocardial infarction (MI) and acute coronary syndrome (ACS). The first biomarkers for MI were identified in the 1950s (Figure 2). Serum glutamic oxaloacetic transaminase, now known as aspartate aminotransferase (AST), was first followed by lactate dehydrogenase (LDH) [14]. While these markers were indeed elevated in acute myocardial infarction (AMI) [15,16,17,18,19], they suffered from a lack of cardiac specificity [20], and the hunt was on to find more specific markers reflecting myocardial insult. This became particularly important with the evolution of MI treatments, including lytic therapy, where directing the right treatment to the right patient became critical. Over time, additional markers have emerged for reflecting heart failure and vascular inflammation, which are discussed below.

Summary data overviewing each biomarker’s indication, reference ranges and the statistical methods used in biomarker development are listed in Table 1. Information for each marker relating to time sampling, time dynamics and non-cardiac causes of biomarker elevations are outlined in Table 2. The recommendations for each biomarker from the AHA/ACC and ESC guidelines, including level of evidence and class of recommendation, is summarized in Table 3. 

### 3.1. Acute Coronary Syndrome (ACS) Biomarkers

#### 3.1.1. Creatine Kinase

Creatine kinase (CK) is an enzyme found in all muscle cells that catalyzes the reaction between adenosine triphosphate (ATP) and creatinine in the production of cellular energy, and it is released into circulation when myocytes are damaged. It was first proposed as a potential biomarker for myocardial infarction in 1960 [21], followed by a study in 1963 comparing 120 patients with AMI to 34 controls in the pre-angiography era [22]. Subsequent studies showed that CK was more specific for coronary occlusion than AST, particularly when looking at patient populations with myocardial damage and hepatic parenchymal damage or acute pancreatitis [23], as well as in patient populations with differing presentations suggestive of AMI, non-AMI heart diseases, and non-cardiac diseases including pulmonary embolism, primary hepatic disease, and cancer [24].

The discovery of isoenzymes of creatine kinase and high concentrations of the cardiac muscle-specific isoform CK-MB in the 1970s [25,26,27] led to the development of assays that confirmed the elevation of circulating CK-MB following myocardial infarction [28,29,30]. Studies examined populations suspected of acute myocardial infarction [31], populations with confirmed or suspected myocardial infarctions, indications for cardiac catheterization or non-cardiac surgery [32], and populations admitted to the acute cardiac unit of Mount Sinai Hospital [33]. In most patients, CK-MB was elevated 4–6 h post-infarction [34,35] and returned to the baseline within 36–48 h [19].

These initial studies into CK and CK-MB primarily reported their results by comparing the means between groups and of the control samples without detailed analyses between groups. These data were reviewed to identify the normal range for CK-MB, and the diagnostic thresholds for elevated CK-MB were defined using the upper limit of the standard [29,36,37,38]. In the 1990s, early diagnosis of MI was improved by assays of sub-forms of CK-MB [34], as reflected by the use of the C-statistic, which has since become the common practice for assessing new biomarkers [35]. Testing of the total CK in the modern laboratory is generally performed by spectrophotometry, with isoform analysis conducted either by electrophoresis or immunoassay [34]. While CK-MB had a long period of clinical utility, it is no longer the biomarker of choice for primary diagnosis of ACS.

#### 3.1.2. Heart Fatty Acid-Binding Proteins (h-FABP)

The search for biomarkers of ACS with improved sensitivity and specificity continued despite the success of CK-MB. Initially described in 1972, fatty acid-binding proteins (FABP) are responsible for the cytoplasmic transport of unsaturated fatty acids in various organs, including the kidneys, myocardium, intestines, and adipose tissues [39]. Heart-type FABP (h-FABP) accumulated interest as a potential biomarker of ACS following observation of its release from ischemic myocardial cells in rats [40].

h-FABP has been recommended as a marker for ACS in patients presenting shortly after symptom onset. Different enzyme-based immunoassays to measure h-FABP found that its plasma levels peaked earlier than CK-MB and LDH, with elevations detectable at approximately 5–10 h following the onset of ACS symptoms [41,42]. Two separate studies investigating patients with acute chest pain and suspected AMI published in 2004 and 2008 found that h-FABP had a higher sensitivity but lower specificity than early troponin T assays for diagnosing AMI within 2 and 4 h of symptom onset [43,44]. The h-FABP concentration reaches its peak 6 h after symptom onset and returns to its baseline level by 24 h [45]. At present, no formal recommendation exists for a clinically useful threshold level for h-FABP. A cut-off of 4 ug/L has been suggested based on calculations of the area under the receiver operating characteristic (ROC) curve, or the C-statistic, in a number of studies [44,46,47].

While high-sensitivity troponin assays remain dominant in current clinical practice (discussed below), there may still be a role for h-FABP in point-of-care testing in primary healthcare, urgent care centers, and emergency departments [47] and as part of a decision tree in combination with high-sensitivity troponins [48]. However, recent studies have found that h-FABP is not a reliable biomarker for ACS [49], even as part of a clinical decision rule with multivariable analyses [50]. More research is required to determine how h-FABP could be meaningfully utilized in a clinical setting.

#### 3.1.3. Troponin

While CK-MB was found to be a reasonable biomarker for AMI, there was a need for markers that could be detected soon after the onset of symptoms which could be utilized either to confirm or refute the diagnosis of AMI, and this led to interest in the troponins. The troponin complex consists of three subunits that work collectively to displace tropomyosin from the cross-bridge binding sites on actin [51,52]. The subunits were first characterized as cardiac proteins in 1965 [51]. Troponin C (TnC) has a role in binding to calcium ions, eliciting a conformational change in troponin I (TnI), which is responsible for inhibiting ATPase [52]. Troponin T (TnT) binds to tropomyosin, and actin-myosin binding and subsequent muscle contraction result. While TnC is present in both skeletal and cardiac muscles, TnT and TnI are specific to cardiac myocytes and remain the most specific biomarkers for myocardial tissue damage.

The first TnI immunoassay was developed in 1987 [53]. This assay detected mean peak TnI levels of 112 ng/mL in patients with AMI and <10 ng/mL in those without 18 h after estimated infarction onset [53]. Similar results were later observed for a TnT immunoassay [54]. Further studies in the 1990s provided compelling evidence for the utility of immunoassays for TnT [55,56,57,58,59] and TnI [60,61], and these were strongly advocated for [38,62].

In the year 2000, experts recommended the upper limit for positive results to be set to the 99th percentile compared with the control population three standard deviations above the population’s mean [63]. Subsequently, the International Federation of Clinical Chemistry advised a control sample size of 300 to increase the precision of the numerous troponin assays available [64]. The current universal definition of AMI utilizes these recommendations [65]. There is still no consensus, however, on what defines a “normal” population in which to determine the 99th percentile [65]. The detection of a rise or fall in the troponin level is necessary for the diagnosis of AMI [65]. This requires serial measurement of troponin levels, but defining a pathological delta is assay-dependent [65]. The first sample should be taken at the time of presentation, and a negative result is routinely followed by repeat testing after 3–6 h, although high-sensitivity assays may allow for a rule-in diagnosis by 2 h [65]. Further measurement after 6 h may be required in ongoing ischemic episodes or in high-risk patients [65].

The last decade has seen continuing improvements in the sensitivity of both troponin I and troponin T, with high-sensitivity immunoassays available in the lab and recently in point-of-care devices [66,67]. Prospective studies have demonstrated the major advantages of the high-sensitivity assays as a part of algorithms for the rapid ruling out of ACS in chest pain patients presenting to the emergency department, helping avoid unnecessary hospitalization without compromising patient safety [68,69,70,71,72,73]. While serial testing strategies have long been the norm, early rule-out pathways using a single high-sensitivity troponin test under a threshold value (<5 ng/L for hsTnI) have been demonstrated to be safe if it has been at least 2 h since symptom onset [74].

However, with the advent of high-sensitivity assays, a greater proportion of the general population have measurable levels of troponin in their blood [75]. Some of these elevations are in an expected range of biological variability, but there is also troponin release after strenuous exercise [76] (including stress testing [77]) and rapid atrial pacing in patients with coronary artery disease [78]). Patients with underlying cardiovascular disease have significantly higher troponin levels at baseline when measured with high-sensitivity assays [79], and a study of apparently healthy participants in the community where increasingly strict selection criteria were applied resulted in progressively lower 99th percentile upper reference limits [80]. A recent study found that baseline high-sensitivity troponin was an independent predictor of coronary heart disease death or pending AMI, even in levels within the normal range [81]. Interestingly, in participants treated with statin therapy, those who had a significant decrease in high-sensitivity troponin levels in response to treatment had the greatest reduction in non-fatal MI and death from coronary artery disease, suggesting a role for high sensitivity troponin in treatment monitoring [81]. This variation within the “normal” range has led to interest in high-sensitivity troponin as a biomarker for CV risk prediction outside the ACS sphere. Advocates highlight that troponin has many features of a good cardiovascular risk biomarker, such as detectably elevated levels in high-risk patients, dynamic reductions in those on risk factor treatment, and additive value to existing risk scoring systems [82]. While there are some challenges, such as the marker’s narrow concentration ranges which correlate with risk, further studies may demonstrate that this is a cost-effective biomarker which could be easily used to supplement existing risk assessment mechanisms.

Outside of these variations within the low-to-normal ranges—which may prove to be useful for non-ACS risk stratification in the future—the major limitation of troponin relates to its inability to distinguish the underlying pathology, leading to troponin elevation. The biomarker is unable to distinguish between an atherosclerotic event in a major epicardial coronary artery versus a type II myocardial infarction, or “MI mimic”, such as myopericarditis or Takotsubo cardiomyopathy. Future efforts to develop more specific markers may provide physicians with a combined tool kit to improve triaging of patients in invasive investigations and avoiding the catheter laboratory for those where percutaneous intervention is not required.

#### 3.1.4. Soluble Lectin-Like Oxidized Low-Density Lipoprotein Receptor-1 (sLOX-1)

Discovered in the late 1990s, the lectin-like oxidized low-density lipoprotein receptor-1 (LOX-1) is a 50-kDa transmembrane glycoprotein which was initially reported as a receptor for oxidized low-density lipoprotein (LDL) [83]. LOX-1 has no binding activity toward native LDL [84], but oxidized or otherwise modified LDL binds to LOX-1 and triggers secretion of chemokines, pro-inflammatory molecules [85], and reactive oxygen species [86], and LOX signaling has been implicated in activation of the NLRP3 inflammasome [87,88]. LOX-1 can be expressed on the surface of endothelial cells, smooth muscle cells, fibroblasts, platelets, and macrophages [89]. LOX-1 expression is undetectable in normal vascular tissue but increases in the presence of atherosclerotic plaque formation [90,91]. The soluble form of LOX-1 (sLOX-1) is released when the extracellular domain of the receptor is cleaved, the specific mechanism of which is still being investigated [84]. Pro-inflammatory signaling via TNF-alpha appears to prime macrophages for sLOX-1 release, however, and this is enhanced by CRP [92] as well as IL-18 [93].

Clinically, sLOX-1 release has been studied as a biomarker for acute coronary syndrome with some significant results. Using enzyme-based immunoassays, sLOX-1 has been found to be elevated in patients who have had any type of ACS [94,95,96], to be raised before any other biomarker [96], and to have better sensitivity and specificity than current biomarkers, including hsCRP [94,96]. There have also been studies showing that sLOX-1 levels are higher in patients with ruptured plaque [97], in more proximal coronary lesions [98], and in more complex lesions [99]. There is also evidence that elevation of sLOX-1 beyond a certain threshold correlates with adverse clinical outcomes following acute coronary syndrome [100,101,102].

The peak elevation of sLOX-1 in STEMI patients occurs earlier and persists for longer (up to 24 h after presentation) compared with other biomarkers of ACS [96]. The diagnostic levels of sLOX-1 still need further evaluation, however, with reports variably using cut-offs of 91.0–131.7 pg/mL [95,96,97] for ACS with plaque rupture, based on analyses of the C-statistic. Additionally, while sLOX-1 levels are elevated earlier than high-sensitivity troponin in ST-elevation myocardial infarction (STEMI) [96], the clinical utility of this is limited, as the majority of patients are diagnosed based on clinical history and ECG findings prior to the return of even the most rapid pathology results. It is yet to be determined whether the specificity of sLOX-1 for atherosclerotic-related events may have benefits in the management of patients with MI-mimics such as myocarditis or Takotsubo cardiomyopathy.

### 3.2. Heart Failure (HF) Biomarkers

#### 3.2.1. Natriuretic Peptides

While both CK-MB and troponin aided the diagnosis of myocardial infarction, it took more time to develop markers for heart failure (HF). A peptide from homogenized rat atrial tissue that exhibited strong natriuretic and chloriuretic effects was described in 1981 [103]. This peptide was named atrial natriuretic factor (ANF), and in 1985, it was postulated that it had a role in the control of blood pressure [104]. Circulating levels of ANF (later renamed to atrial natriuretic peptide (ANP)) were found to be elevated in heart failure [105], and it became an important diagnostic and prognostic biomarker [106]. The N terminus of pro-ANP (NT-ANP) is the inactive portion of the prohormone and is cleared from circulation more slowly, allowing for easier detection in the serum [106]. Subsequently, a novel compound in porcine brains was identified that exhibited similar effects to ANF, aptly named brain natriuretic peptide (BNP) [107]. It was later found that BNP was also present in the hearts of both pigs [108] and rats [109]. The amino acid sequence of human BNP was rapidly identified [110], and shortly afterward, a radioimmunoassay was developed, identifying BNP in human heart samples and revealing the ventricles as the major site of BNP secretion [111].

Initial studies showed that the levels of BNP in plasma were raised in patients with HF [111] and that the level of BNP increased with the severity of heart failure [111,112]. BNP was found to have a higher specificity and positive predictive value for a diagnosis of HF than ANP [113]. A subsequent systematic review found BNP to be a more accurate diagnostic marker for heart failure than NT-ANP [114]. A preliminary assessment of healthy subjects showed that the mean BNP level (±SD) was 1.8 ± 1.0 pmol/L and that in 8 subjects with HF, the median BNP level was 30.5 pmol/L [115].

This was followed up by larger cohort studies [116,117,118], which confirmed significantly elevated levels of BNP in patients with systolic dysfunction and identified the strong negative predictive value of a low BNP result. However, BNP’s elevation in other conditions such as atrial fibrillation [117] was also identified, and thus echocardiography is still considered required for the diagnosis of HF [119].

BNP is cleaved from the C-terminal end of its prohormone, pro-BNP. The N-terminal fragment, N-terminal pro-BNP (NT-proBNP), is also released into circulation [120]. NT-proBNP has been found to be comparable to BNP in patients with an impaired left ventricle ejection fraction [121,122]. NT-proBNP results have been shown to be valuable for diagnosis of HF in patients presenting acutely with dyspnea [123,124,125,126], as well as correlating with prognostic outcomes in acute and chronic HF [126,127,128,129,130]. Both BNP and NT-proBNP are detectable using rapid immunoassays, but the potential advantages of NT-proBNP include its greater range of values [131] and longer half-life [132]. BNP and NT-proBNP measures are strongly correlated (correlation coefficient of 0.81), and in patients with HF with a reduced ejection fraction, the median ratio of NT-proBNP/BNP was 6.25/1, which was consistent across BNP deciles [133]. Interestingly, this ratio was found to be significantly higher in patients of an older age, male sex, higher creatinine, and atrial fibrillation and lower in those with obesity and a history of myocardial infarction [133]. Further research into whether different cut-offs should be used in certain clinical conditions (e.g., atrial fibrillation or renal impairment) is ongoing. Recent Australian guidelines recommend clinical use of natriuretic peptides, describing strong evidence that they are useful in diagnosing suspected HF but only weak evidence that they have utility as prognostic factors [134]. A plasma BNP level of <100 ng/L or an NT-proBNP level of <300 ng/L exclude heart failure [134]. Rigid cut-offs for positive BNP and NT-proBNP results, however, are limited in accuracy due to the multiple factors that influence natriuretic peptide levels, including age, kidney function, the presence of atrial fibrillation, sex, and weight [134]. For these reasons and others, the ESC guidelines use the test as a rule-in test rather than a rule-out test, including BNP ≥ 35 pg/mL or NT-proBNP ≥ 125 pg/mL as part of the diagnostic algorithm for heart failure [135]. Sex-specific differences in natriuretic peptide levels have been detected, but these differences have not been included in diagnostic or risk prediction models [136].

BNP has also been shown to have clinical utility in guiding management of heart failure, particularly regarding fluid volume overload. The utility of BNP and NT-proBNP in guiding management of HF is still unclear, with some evidence suggesting biomarker-guided therapy reduces mortality [137] and some evidence finding no difference compared with symptom-guided management [138,139].

#### 3.2.2. Galectin-3

There was significant interest in galectin-3 as a biomarker when it was first implicated in HF in 2004, when it was found to be specifically overexpressed in the myocardium of Ren-2 rats that would go on to develop HF [140]. Additionally, it was demonstrated that galectin-3 promoted cardiac fibroblast proliferation and increased collagen I deposition, leading to fibrosis [140]. Galectin-3 was found to be reliably measured in enzyme-based immunoassays which do not cross-react with other members of the galectin family [141].

Initial clinical studies showed that while galectin-3 lacked utility for HF diagnosis, galectin-3 was an independent prognostic factor for all-cause mortality and HF hospitalization, suggesting it had use as a prognostic biomarker [141,142,143]. A major study indicated that the baseline galectin-3 measurements were sufficient, with repeat levels after 6 months granting no further prognostic information [141].

Multiple studies on galectin-3 in HF followed. A 2017 meta-analysis reviewed 18 studies with a total of 32,350 patients [144]. After adjustment for age, sex, BNP, renal function, and diabetes, they calculated hazard ratios (HRs) of 1.10 (95% CI: 1.05–1.14) for all-cause mortality, 1.22 (95% CI: 1.05–1.39) for cardiovascular related mortality, and 1.12 (95% CI: 1.04–1.21) for risk of HF with each 1 standard deviation rise in galectin-3 concentrations. In specific analysis for cardiovascular mortality, they calculated an HR of 1.44 (95% CI: 1.09–1.79) for HF patients for every 1 standard deviation increase in galectin-3 concentrations.

Other studies have typically analyzed dichotomous outcomes using a cut-off value of 17.8 ng/mL for galectin-3 when assessing HF, with a significantly increased risk of rehospitalization and mortality due to HF when levels exceeded 17.8 ng/mL [144,145], calculated using the C-statistic. Several prospective studies support these findings [146,147,148].

In addition to its use as a biomarker for HF prognosis, reports have increasingly suggested that galectin-3 may also provide a possible therapeutic target in HF due to its role in the development of myocardial fibrosis and diastolic dysfunction [149,150,151]. Further research into this area is ongoing.

However, due to some limitations, the use of galectin-3 in clinical care of patients with or at risk of heart failure is not routine. Galectin-3 levels are elevated in a number of other fibrotic diseases, including liver cirrhosis [152,153] and pulmonary fibrosis [154], as well as in patients with renal insufficiency [155]. These findings of elevation in other fibrotic processes reduce its specificity as a cardiac biomarker. Furthermore, the effect of age on galectin-3 levels is inconclusive thus far [144].

#### 3.2.3. Suppression of Tumorigenesis-2 (sST2)

Suppression of tumorigenesis-2 (ST2), first reported in 1989 [156], is a receptor related to IL-1 with two main isoforms: transmembrane or cellular (ST2L) and soluble or circulating (sST2) [157]. IL-33 is the ligand of ST2 [158], which is secreted by most cells in response to damage [159], including myocardial stress [160]. The IL-33/ST2 system is cardioprotective and reduces myocardial fibrosis, prevents cardiomyocyte hypertrophy, reduces apoptosis, and improves myocardial function [157,161,162]. sST2, however, is thought to be a decoy receptor that sequesters IL-33, negating its cardioprotective and antihypertrophic effects [157,161]. sST2 concentrations measured by an enzyme-based immunoassay are elevated in patients with AMI and acute HF and correlate with the infarct size and cardiac dysfunction [163], hemodynamic decompensation [164], and risk of death [165].

sST2 has generated interest as a novel biomarker in both acute and chronic HF [166,167]. sST2 concentrations have been found to be higher in chronic HF than healthy patients [168], and sST2 is thought to be a prognostic marker for worse outcomes [169]. A recent meta-analysis has found that sST2 is a predictor of both all-cause and CV death in chronic HF outpatients [170]. In acute HF, sST2 concentrations are higher [171], and sST2 was found to be the strongest predictor of death at 1 and 4 years [165], with similar prognostic capabilities seen in patients with HF with preserved ejection and with a reduced ejection fraction [172].

However, baseline sST2 concentrations have failed to predict incident HF or cardiovascular events, but they do predict all-cause mortality [173]. sST2 was found in one large study to be the biomarker that added the most prognostic value to clinical risk models, predicting short-term and 1-year mortality in acute decompensated HF [174], using the difference in the C-statistic as well as computing the net reclassification improvement (NRI) and integrated discrimination improvement (IDI). sST2 may also play a role in monitoring therapy for HF [166]. sST2 levels have been observed to decrease in patients treated with beta blockers [175] and valsartan [176], and spironolactone treatment was beneficial in patients with elevated sST2 [177].

Clinical use of sST2 is increasing. Measures of biomarker utility including calibration, reclassification, and discrimination have been improved with the addition of sST2 to existing models [178,179,180,181]. The 2017 American Heart Association (AHA) guidelines support the use of an sST2 assay for prognosis and risk stratification in CHF patients, with a Class IIb, level of evidence: B-NR (non-randomized) recommendation [182].

### 3.3. Atherosclerosis Risk

#### High-Sensitivity C-Reactive Protein (hsCRP)

The focus of most early biomarker work was related to the diagnosis of AMI and heart failure, but there has been a paucity of markers for atherosclerosis and coronary artery disease itself. Instead, physicians and their patients rely on risk factor algorithms to assess who may be at high risk of an acute cardiovascular event. This is an area that is likely to be transformed over the next decade. Inflammation is known to play a key role in atherosclerosis. As such, investigators have dedicated substantial efforts to identifying a potential role for high-sensitivity C-reactive protein (hsCRP) measures. This sensitive enzyme-based immunoassay has been developed to be capable of detecting low levels of the molecule CRP, which is an acute phase reactant released by the liver in response to inflammation. This allows discrimination between values that were previously considered to be in the “normal range” using the older, less-sensitive methodology. Measurements of CRP have been tentatively linked to cardiac health since the 1980s, with studies finding that CRP levels were raised in patients who had been diagnosed with AMI or angina [183,184].

Baseline CRP measurements have been found to be higher in men who went on to have an AMI (1.51 vs. 1.13 mg/L) or stroke (1.38 vs. 1.13 mg/L) compared with men who did not, and the risk of AMI increased with each quartile increase of CRP levels [185]. These findings were expanded upon in post-menopausal women, and hsCRP was found to correlate most strongly with cardiovascular events compared with 11 other markers [186].

Increased baseline CRP results have been found to correlate independently with subsequent risk of cardiovascular events in a number of prospective trials in the late 1990s [187,188]. The risk increase associated with elevated CRP in the Framingham study remained statistically significant across CRP quartiles, independent of other risk factors [188].

More recent studies have examined whether identifying CRP levels would be a useful tool for clinical and pharmacological management. The JUPITER study examined the impact of statins on lowering hsCRP levels and future cardiovascular events. Statin therapy led to a reduction in levels of hsCRP in some participants, which in turn was associated with a reduced incidence of cardiovascular events [189]. In addition to inhibiting HMG-CoA reductase, statins are thought to inhibit the synthesis of isoprenoid intermediates of the mevalonate pathway, which may account for other proposed benefits of statins, including improvements in endothelial function, plaque stabilization, and reduced inflammation, as demonstrated by the reduction in hsCRP [190,191].

The American Heart Association and United States Center for Disease Control have defined a normal level (low-risk) hsCRP as <1.0 mg/L, a moderately increased risk level as 1–3 mg/L, and high-risk levels as >3.0 mg/L [192,193] based on the approximate levels in the general population [192]. Various studies have categorized hsCRP results into tertiles [194,195], quartiles [196], and quintiles [197] and computed the C-statistic [197]. While these data have proven convincing enough for hsCRP to now appear in many guidelines, hsCRP is not yet widely used in cardiovascular risk stratification models.

**Table 1 cells-11-00588-t001:** Summary of biomarker indications, reference ranges, and statistical approaches used. Abbreviations: ACS = acute coronary syndrome; AST = aspartate aminotransferase; AMI = acute myocardial infarction; LDH = lactate dehydrogenase; CK = creatine kinase; CK-MB = myocardial creatine kinase isoenzyme; h-FABP = heart-type fatty acid-binding protein; TnI = troponin I; TnT = troponin T; BNP = brain natriuretic peptide; hsCRP = high-sensitivity C-reactive protein; sLOX-1 = soluble lectin-like oxidized low-density lipoprotein receptor-1; sST2 = soluble suppression of tumorigenesis-2.

Biomarker	Indication	Reference Range	Statistics
CK-MB	AMI	>99th percentile of upper reference limit of sex-specific controls for assay [65] >20–25 U/I [30,37] >30 U/I [29] CK-MB_2_:CK-MB_1_ ratio > 1.5 [34,35]	Analysis of frequency distribution [29], C-statistic [35]
Troponin	AMI/CV Risk	>99th percentile of upper reference limit for assay [65]	≈3 SDs above the mean for normal range [63]
sLOX-1	AMI	>91.0–131.7 pg/mL (suggested) [95,96,97]	C-statistic [95,96,97]
h-FABP	AMI	>4 ug/L [46,47]	C-statistic [43,44,47]
BNP	HF	Rule out: <100 ng/L [134]	C-statistic [114,116]
		Rule in: >400 ng/L [134]	
NT-proBNP	HF	Rule out: <300 ng/L [134]	C-statistic [198,199]
		Rule in: age < 50; >450 ng/L [134] Rule in: age 50–75; >900 ng/L [134] Rule in: age > 75; >1800 ng/L [134]	
Galectin-3	HF	>17.8 ng/L [144,145]	C-statistic [144,145,146]
sST2	HF	>35 ng/L [200,201]	C-statistic [202], NRI, and IDI [174]
hsCRP	CV Risk	High risk: >3 mg/L [192,193]	Tertiles [194,195], quartiles [196], quintiles [197], and the C-statistic [197]
		Increased risk: >1 mg/L [203]	

**Table 2 cells-11-00588-t002:** Time sampling, time dynamics, and non-cardiac causes of altered levels of cardiac biomarkers. Abbreviations: AF = atrial fibrillation; AKI = acute kidney injury; ALD = alcoholic liver disease; ARNI = angiotensin receptor-neprilysin inhibitor; BNP = brain natriuretic peptide; CK-MB = myocardial creatine kinase isoenzyme; COPD = chronic obstructive pulmonary disease; DM = diabetes mellitus; HF = heart failure; h-FABP = heart-type fatty acid-binding protein; hsCRP = high-sensitivity C-reactive protein; HTN = hypertension; LVH = left ventricular hypertrophy; NAFLD = non-alcoholic fatty liver disease; NT-proBNP = N-terminal pro-BNP; PE = pulmonary embolism; sLOX-1 = soluble lectin-like oxidized low-density lipoprotein receptor-1; sST2 = soluble suppression of tumorigenesis-2.

Biomarker	When to Take Sample (Time Sampling)	Biomarker Changes over Time (Time Dynamics)	Non-Cardiac Causes of Altered Levels
CK-MB	4–6 h after symptom onset [34,35]	Peak occurs after 16–30 h, returns to baseline by 24–36 h [204]	Elevated in skeletal muscle injury [205], vigorous exercise [206], stroke [207], trauma patients [208], and kidney disease [209,210]; 1.2–2.6x higher 99th percentile in males [211] and post-operatively in spinal surgery [212]
h-FABP	2–4 h after symptom onset [43,44]	Peak occurs 6 h after symptom onset, returns to baseline by 24 h [45]	Elevated in AKI [213], PE [214], stroke [215], sepsis [216], acute HF [217], NAFLD [218], smoking, and COPD [219]
Troponin	At presentation and then 2–6 h later if the first result is negative [65]	Peak occurs at 12–48 h [220], returns to baseline by 14 days [221]	Elevated in sepsis [222], critical illness [223], LVH [224], coronary vasospasm [225], stroke [226], AF [227], heart failure [228], myocarditis [229], dialysis patients [230], males, black people, DM, and HTN [231]; lower in smoking, alcohol use, and statin use [231]
sLOX-1	At presentation [96]	Peak is maintained from presentation up to 24 h [96]	Conflicting association with smoking [95,232] and not significantly correlated with lipids, diabetes, or hypertension [94,95]
BNP	At presentation for acute dyspnea [116] as a screening tool [117,118,233]	Levels remain elevated in untreated HF; treatment may lower levels to normal range [234,235]	Elevated in smokers [236] and renal insufficiency [237,238]; lower in obesity [239], even in patients with HF [240]; degraded by neprilysin (ARNI therapy causes BNP elevation) [241]
NT-proBNP	At presentation for acute dyspnea [123,124,125,126]	Levels remain elevated in untreated HF; treatment may lower levels to normal range [234]	Elevated in smokers [242]; renal insufficiency (greater than BNP) [243]. Lower in obesity [239]. Not degraded by neprilysin (can be used to monitor ARNI therapy) [241].
Galectin-3	At presentation as a prognostic marker [144]	Levels remain stable over time [141]	Conflicting evidence for association with sex, age, DM, and HTN [244,245]
sST2	Serially, as a prognostic marker [172,201,246]	Levels may remain elevated (indicating worse prognosis) or decrease by 48–72 h [247]	Elevated in smoking [248], males, DM [249], and ALD [250]
hsCRP	As a risk-enhancing factor at screening for patients at borderline or intermediate risk of atherosclerotic CVD [251]	Levels may fluctuate considerably over time [252], and statin therapy may reduce levels [189]	Elevated in smoking [248,253] and other inflammatory processes [254,255]

**Table 3 cells-11-00588-t003:** Biomarker recommendations from AHA/ACC and ESC guidelines. Abbreviations: AHA = American Heart Association; ACC = American College of Cardiology; ACS = acute coronary syndromes; BNP = brain natriuretic peptide; B-NR = level of evidence B (non-randomized trials); B-R = level of evidence B (randomized trial); CK-MB = myocardial creatine kinase isoenzyme; COR = class of recommendation; CVD = cardiovascular disease; ESC = European Society of Cardiology; HF = heart failure; h-FABP = heart-type fatty acid-binding protein; hsCRP = high-sensitivity C-reactive protein; hscTn = high-sensitivity cardiac troponin; LOE = level of evidence; NT-proBNP = N-terminal pro-BNP; sLOX-1 = soluble lectin-like oxidized low-density lipoprotein receptor-1; sST2 = soluble suppression of tumorigenesis-2. Classes of recommendation: I = use is recommended; II = use is reasonable or should be considered; IIa = use is reasonable; IIb = use may be reasonable; III = use may be considered. Levels of evidence: A = data derived from multiple randomized, controlled trials or meta-analyses; B = data derived from a single randomized clinical trial (B-R) or large non-randomized studies (B-NR); C = consensus opinion of experts, case studies, registries, or standard of care. Left blank if data not listed in guideline.

Biomarker	AHA/ACC			ESC		
	Recommendation	COR	LOE	Recommendation	COR	LOE
CK-MB	Not recommended for diagnosis of ACS [256]	III	A	Not recommended for diagnosis of ACS [257]	III	
h-FABP	Not in guidelines			Not recommended for diagnosis of ACS [257]	III	B
Troponin	Diagnosis of ACS [256]	I	A	Diagnosis of ACS [257]	I	B
	Additive risk stratification in chronic HF (hscTn) [182]	IIb	B-NR			
sLOX-1	Not in guidelines			Not in guidelines		
BNP and NT-proBNP	Screening for HF [182]	IIa	B-R			
	Diagnosis of HF [182]	I	A	Diagnosis of HF [135]	I	B
	Prognosis or disease severity in chronic HF [182]	I	A			
	Prognosis in ADHF [182]	I	A			
	Pre-discharge for prognosis [182]	IIa	B-NR			
Galectin-3	Additive risk stratification in chronic HF [182]	IIb	B-NR	Not in guidelines		
sST2	Additive risk stratification in chronic HF [182]	IIb	B-NR	Not in guidelines		
hsCRP	As a risk enhancing factor to aid discussion of statin therapy initiation [258]			Not recommended for risk stratification in CVD prevention [259,260]	III	B

## 4. Prospective Biomarker Trials

While biomarkers can be validated in retrospective cohort or clinical trials, the most convincing evidence for the clinical utility of biomarkers in the future will come from randomized controlled trials (RCTs) comparing health outcomes between patients receiving novel biomarker-guided treatment and the standard of care. This applies to diagnostic and prognostic markers, as well as those that may guide ongoing treatment. There are three main designs of RCTs assessing biomarkers: biomarker-stratified designs, enrichment designs, and biomarker strategy designs [2,261].

In biomarker-stratified trials, patients are analyzed by biomarker-defined groups in the context of a given therapy where there is no pre-specified knowledge of a benefit to the biomarker-defined group. Biomarker assessment can either be performed prospectively or retrospectively, though the prospective study design has the benefit of ensuring adequate distribution of patients in the treatment arms and is a true representation of biomarker-guided treatment. However, most examples of this design in the cardiovascular sphere are from retrospective analyses, such as in the CORONA trial [262]. CORONA randomized older patients with heart failure who were not clinically felt to require lipid-lowering therapy to either rosuvastatin or a placebo, and the overall results of the study demonstrated no significant improvement in any of the primary outcomes. In secondary analysis of the trial [263], patients with galectin-3 levels lower than 19.0 ng/mL were shown to have benefitted more from rosuvastatin therapy, identifying a potentially useful subgroup for precision therapy that could not have been predicted prior to the trial.

In contrast, enrichment designs are applied to demonstrate that a biomarker-defined subgroup of patients benefits from a particular therapy and that they are useful when evidence suggests that treatment of patients in a particular population would be beneficial [2]. This design was utilized in the GRAVITAS trial, which studied the effect of high-dose clopidogrel in post-PCI patients with high residual platelet reactivity on standard-dose clopidogrel (though no benefit was seen with the higher dose in patients with high residual platelet activity) [264] and in the CANTOS [265] trial, which studied the effect of antibody-blocking interleukin-1β on patients with previous ACS who had higher levels of high-sensitivity CRP, demonstrating a significantly lower rate of recurrent cardiovascular events when compared with a placebo.

Finally, biomarker strategy designs are used to evaluate the value and clinical utility of applying a biomarker in clinical care and to assess the impact it has on management. This is accomplished by randomizing patients either to a biomarker-guided treatment arm or to a control arm where they receive the usual care. Examples of this include the PROTECT trial [266], where NT-proBNP-guided therapy was found to be superior to standard care, the ongoing GUIDE-IT trial [139], where NT-proBNP levels would be used to guide the intensity of the therapy, and in the ongoing SCOT-HEART [267] trial, where CT coronary angiography (as an “imaging biomarker”) was being used to stratify the treatment of stable chest pain.

These study designs highlight that biomarkers have significant potential to improve the precision of risk stratification and of treatment directions for patients in the future. While biomarker discovery and validation has, to a large extent, focused on the performance of the biomarker in the general population, moving forward, we expect to see more studies that seek to apply biomarkers to subgroups in the population where traditional risk markers fail to adequately assess risk or diagnose disease. These types of studies lead into the concept of precision medicine and are starting to pave a new path in biomarker development strategies.

The incorporation of biomarkers into trial design and outcome analysis will continue to expand and be refined as our repertoire of biomarkers and our understanding of what they reflect in the underlying patient improves with time.

## 5. Conclusions

The history of biomarkers in cardiovascular disease is wide-ranging, and while it incorporates several markers critical to the practice of cardiology such as troponin and NT-pro-BNP, it also includes several which were found to have limitations which kept them consigned to the benchtop. Whether we will see more inclusion of these markers in the diagnosis of cardiac pathologies in the future remains to be seen. There are still significant areas of unmet need in the cardiovascular biomarker space, including markers for detection of early atherosclerosis that would enable more aggressive risk factor management prior to a cardiac event and better ways to definitively diagnose STEMI from mimicking conditions such as myocarditis and Takotsubo cardiomyopathy, allowing avoidance of unnecessary invasive angiography.

As discussed above, most cardiac biomarker evaluations to this point have been dichotomous. There remains room to improve on this model, with biomarkers classifying patients into risk and severity groups and aligning them with individualized treatments. Furthermore, individualized weighting of factors in combined risk models (e.g., clinical findings, laboratory results, and imaging) may become standard practice as our ability to identify biological differences improves with the technology.

Looking to the future, there are exciting opportunities for rapid progress to be made in cardiac biomarker research as we gain more knowledge of individual differences in biology and the consequent personalization of cardiovascular medicine. The field of cardiology would benefit from markers that could inform about more individualized pathologies and responses to treatment, such as markers that identified the degree of vascular damage in a specific patient from hypertension or, conversely, one that showed beneficial changes in response to medical therapy or increases in exercise. Advancements in technology will allow us to better identify biomarkers using unbiased research strategies which allow us to widely survey the human proteome, metabolome, and immunophenotype, identifying markers or signals which have an association with cardiovascular diseases. The addition of a biosignature to the currently used clinical risk profile with imaging measures may significantly improve our ability to diagnose those at substantial risk of cardiovascular disease. Ongoing efforts to design and undertake innovative clinical trials will be necessary to demonstrate the utility of current and future cardiac biomarkers, providing crucial clinical information to allow integration into current practice.

## Figures and Tables

**Figure 1 cells-11-00588-f001:**
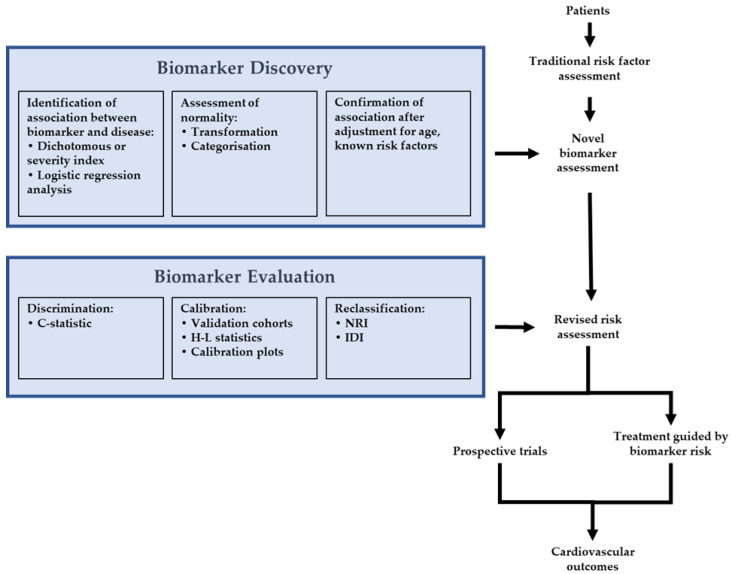
Framework for biomarker discovery, evaluation, and incorporation into clinical trials. Abbreviations: H-L = Hosmer–Lemeshow; NRI = net reclassification index; IDI = integrated discrimination improvement.

**Figure 2 cells-11-00588-f002:**
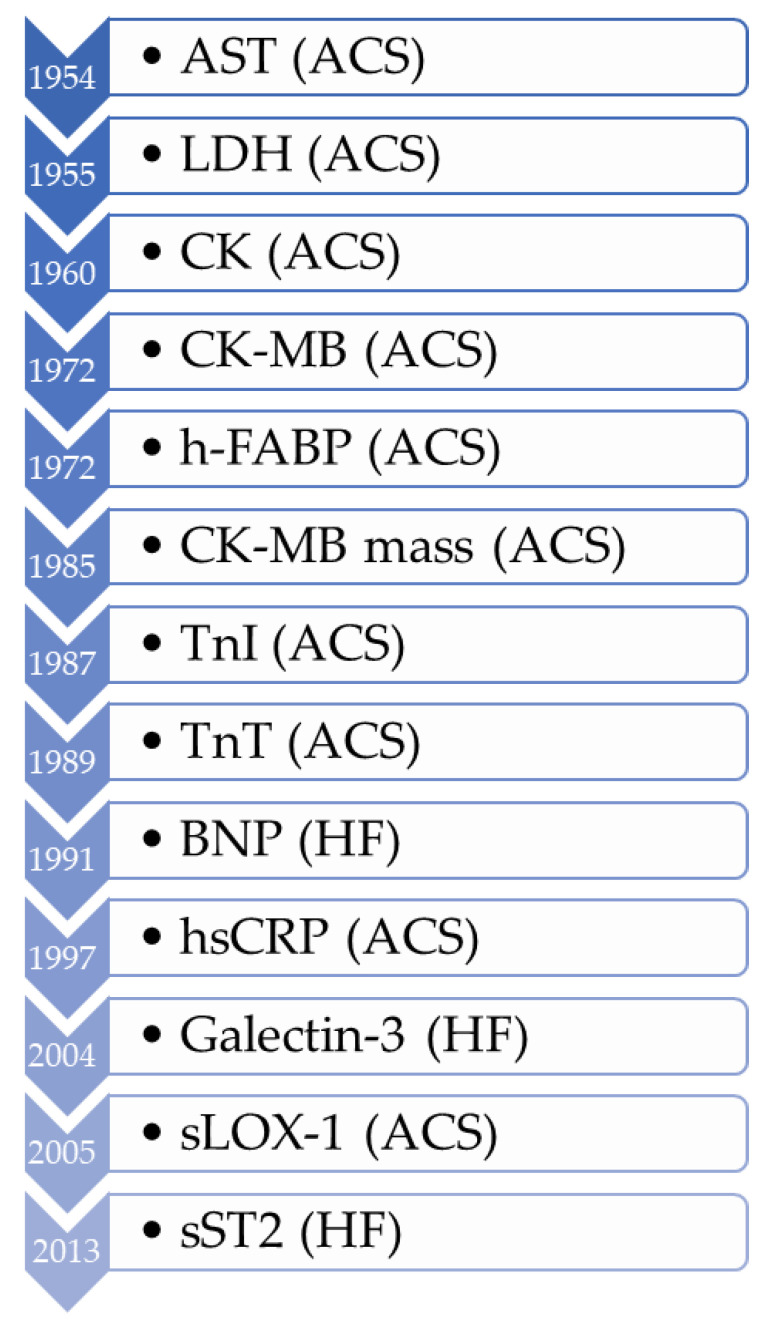
Timeline of biomarker development in cardiology. Abbreviations: AST = aspartate aminotransferase; ACS = acute coronary syndromes; LDH = lactate dehydrogenase; CK = creatine kinase; CK-MB = myocardial creatine kinase isoenzyme; h-FABP = heart-type fatty acid binding protein; TnI = troponin I; TnT = troponin T; BNP = brain natriuretic peptide; hsCRP = high-sensitivity C-reactive protein; sLOX-1 = soluble lectin-like oxidized low-density lipoprotein receptor-1; sST2 = soluble suppression of tumorigenesis-2.

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
