# Peer review of "Biomarker Development in Cardiology: Reviewing the Past to Inform the Future"

_cells, 2022, doi:10.3390/cells11030588_

Round 1
Reviewer 1 Report
This review article analyses the development and use of biomarkers in cardiology. The first part of the text focuses on the statistical methodology used to assess possibilities of a potential biomarker to be used in the diagnostic algorithms, grading diseases severity, as well as the prognostic possibility in various CV diseases. In addition, some of the biomarkers have been considered as potentially important for the introduction and monitoring of certain therapies and procedures.
The paper is well written, easy to read and follow. It has very good tables and references.
Minor suggestions:
- In table 1 for each of the biomarkers analyzed in the text, the indications in which their determination is recommended, references for the ranks of normal values, and the values of diagnostic ranks are stated. It would be useful to put recommendations for the specific methodology optimal to determine individual biomarkers (the most reliable methodology). This might help to more easily compare the results.
- Generally, all listed biomarkers should be given either in a separate table or in the text, data on the time dynamics of biomarker changes in concentration for specific conditions/diseases as well as recommendations when to do blood sampling (time sampling) in order to measure values for each biomarker useful for diagnosing, prediction or make decisions to guide and monitor therapeutic effects.
- No data have been provided on the use of biomarkers in certain diseases and conditions based on the recommendations of the current official guidelines issued by relevant international societies (for example European Society of Cardiology, ACC etc). It would be useful to put in a table or in a text the recommendation (issued by specific association), level of recommendation, and level of evidence. These recommendations are based on high-quality studies, expert opinions, or meta-analyzes. This is important because it would provide a clear insight into which biomarker is important for the specific clinical scenario (clinical algorithm).
- It might be useful to list (in a table) for each biomarker, which condition may affect its blood level (physiological conditions, drugs, habits such as smoking, alcohol, physical activity, high temperatures, etc… This info might be clinically useful in order to avoid sampling and determination of individual biomarkers in conditions and situations when the findings are unreliable and when is difficult to assess the clinical significance of the obtained results
Reviewer 2 Report
This review discussed clinical significance and development of cardiac biomarkers and also discussed statistical approaches to assessing new biomarkers. The manuscript was well written and the discussion is almost fair. I have several comments on this paper.
- Although a lot was discussed about BNP, discussion regarding ANP was scarce. It would be better to discuss ANP.
- The authors discussed statistical approaches to assessing new biomarkers in section 2 (statistical Approaches to Assessing New Biomarkers) and contents discussed in section 2 is meaningful. However, the main theme of this review is clinical significance and development of cardiac biomarkers and section 2 somewhat missed the point. It would be better for section 2 to be deleted or shortened.
Reviewer 3 Report
The article is interesting and reads well . However , in the manuscript more emphasis should be given to established biomarkers in any covered area ( Acute Coronary Syndrome , Atherosclerotic Risk , Heart Failure etc.. ) used routinely in clinical practice ,at the end including a subchapter on novel promising biomarkers . For instance , in the Acute Coronary Syndrome chapter ,I would suggest not to include a subchapter dedicated to with h-FABP ( which is not routinely used ) , but one dealing with a series of biomarkers , including , besides h-FABP , copeptin and other markers linked to plaque activation and instability ( metalloproteinase-9 , P selectin , soluble CD-40 ligand etc.. ) . The same considerations apply to the Heart Failure biomarkers chapter . It is advisable to underline the differences between NTproBNP and BNP levels in various clinical scenarios ( see DOI: 10.1161/CIRCHEARTFAILURE.119.006541 ) . In the troponin subchapter , protocols to “rule-in” “rule-out” myocardial infarction should be discussed , in particular single versus multiple samples strategies. The possible role of troponins for cardiovascular risk stratification in the general population should be better focused , please refer to doi:10.1093/eurheartj/ehaa083.Author Response
Please see the attachment.

Round 2
Reviewer 3 Report
In the revised version the authors improved the quality of their manuscript